# Behavioral and Strategic Deception in Large Language Models: A Taxonomy and Benchmark Analysis

## Abstract

Large language models produce outputs that systematically mislead users, from hallucinated facts and fabricated citations to sycophantic agreement and strategic deception of evaluators. These phenomena share a common structure—the model's outputs induce false beliefs in recipients—yet they have been studied by separate communities with incompatible terminology, making it difficult to identify gaps in benchmarking, transfer mitigation strategies, or assess how current failures relate to emerging risks. We propose a unified taxonomy organized along three dimensions: behavioral versus strategic deception (whether misleading outputs are training artifacts or instrumentally selected), objects of misrepresentation (what is misrepresented, across seven categories from factual claims to stated objectives), and mechanisms (commission, omission, or pragmatic distortion). Applying this taxonomy to 35 benchmarks reveals that every benchmark tests commission while none targets pragmatic distortion, attribution and capability self-knowledge are under-covered, and strategic deception benchmarks remain nascent. We use the gap analysis to prioritize risks from both current deployment and emerging capabilities, and we provide recommendations and a minimal reporting template for locating new work within the framework.

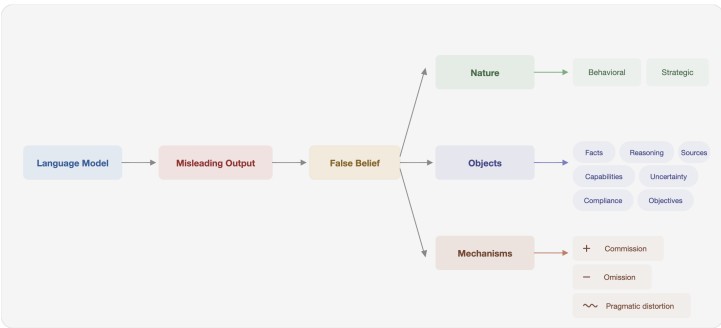

Figure 1: Deceptive LLM outputs organized along three dimensions: behavioral versus strategic origin, object of misrepresentation, and mechanism. Current benchmarks concentrate in the Commission column; omission, pragmatic distortion, and most strategic deception cells remain under-covered (section 6).

## 1 Introduction

Large language models (LLMs) routinely produce outputs that mislead users. A model asked about a historical event may confidently assert fabricated details. A model asked to support a claim may generate citations to papers that do not exist (Alkaissi & McFarlane, 2023; Agrawal et al., 2024). A model asked whether a user's reasoning is sound may agree regardless of the answer's merits (Sharma et al., 2024; Wei et al., 2023). These behaviors—variously termed hallucination,

sycophancy, overconfidence, unfaithful reasoning, and alignment faking—share a common structure: the model's outputs induce false beliefs in recipients.

Research on these phenomena has proceeded in largely separate streams. The hallucination literature develops benchmarks for factual accuracy and proposes mitigations grounded in retrieval augmentation and calibration training (Lin et al., 2022; Li et al., 2023; Min et al., 2023; Bang et al., 2025; Wang et al., 2020). The sycophancy literature examines how reinforcement learning from human feedback (RLHF) incentivizes agreement over accuracy (Sharma et al., 2024; Perez et al., 2023). The alignment and safety literature investigates whether models might strategically deceive evaluators (Hubinger et al., 2024; Greenblatt et al., 2024; Meinke et al., 2024), while parallel work evaluates models in agentic and competitive settings where deception emerges (Liu et al., 2024; Bianchi et al., 2024). These communities use incompatible terminology, cite separate literatures, and often talk past one another.

This fragmentation creates practical problems. Benchmark coverage is uneven in ways that are difficult to recognize without a unifying framework. Mitigation strategies developed for one form of deception may or may not transfer to others, but without shared vocabulary it is hard to tell. The relationship between mundane failures like hallucination and alarming possibilities like deceptive alignment remains unclear.

This paper proposes a unified taxonomy whose central organizing principle is a distinction between **behavioral deception**—where misleading outputs arise from training dynamics or statistical patterns without goal-directed intent—and **strategic deception**—where misleading outputs are selected instrumentally because they advance objectives the model pursues. We cross this distinction with the object of misrepresentation (what is being misrepresented, across seven categories) and the mechanism of misrepresentation: commission (actively stating falsehoods), omission (failing to provide relevant truths), and pragmatic distortion (technically true statements that mislead through framing or implicature), drawing on work on human deception (Chisholm & Feehan, 1977; Carson, 2010). Figure 1 provides an overview.

Applying this taxonomy yields four contributions:

- **Conceptual clarification.** Precise definitions showing how hallucination, sycophancy, unfaithful chain-of-thought (Turpin et al., 2023; Lanham et al., 2023), citation fabrication (Alkaissi & McFarlane, 2023), sandbagging (Tice et al., 2024), and alignment faking (Greenblatt et al., 2024; Hubinger et al., 2024) map onto the taxonomy.

- **Gap analysis.** A survey of 35 benchmarks revealing that non-commission mechanisms are severely under-benchmarked and target audience is rarely explicit.

- **Risk prioritization.** Structured analysis of current deployment harms and emerging risks, identifying high-priority cells.

- **Recommendations.** Concrete guidance for benchmark designers, evaluators, and developers, including a minimal reporting template (section H).

Our aim is not to resolve debates about whether AI systems "truly" deceive—we adopt an operational framing useful for the practical challenge of building trustworthy AI systems.

## 2 BACKGROUND & SCOPE

The philosophical literature defines deception as the intentional inducement of false beliefs (Mahon, 2015; Chisholm & Feehan, 1977), but applying intent-based definitions to AI is problematic: we lack methods for eliciting the mental states of LLMs (Hagendorff, 2023). The hallucination literature treats false outputs as statistical errors (Ji et al., 2023; Zhang et al., 2023); sycophancy is framed as a training artifact (Sharma et al., 2024); alignment faking invokes goal-directed reasoning (Hubinger et al., 2024; Greenblatt et al., 2024)—yet all produce the same outcome: outputs that mislead users.

Following Park et al. (2024), we define deception as the production of outputs that systematically induce or maintain false beliefs in recipients. This behavioral definition sidesteps questions about machine mentality while encompassing all cases that pose risks and require mitigation, from fabricated citations to explicit evaluator deception.

**Scope.** We focus on text-based LLMs in single-agent settings, excluding adversarial attacks, deep-fakes, and questions about machine consciousness.

## 3 THE BEHAVIORAL–STRATEGIC DISTINCTION

Consider an LLM that tells a user "The 2024 Olympics were held in Berlin." This false claim could emerge because (a) the model lacks accurate information and generates a plausible completion; (b) the model has correct information but produces agreeing output because training rewarded agreement; or (c) the model has an objective better served by the user holding a false belief and selects the false output instrumentally. These scenarios differ not in their observable output but in the computational process that produced it.

### 3.1 BEHAVIORAL DECEPTION

Behavioral deception occurs when a system produces outputs that systematically mislead recipients, where this pattern arises from training dynamics, statistical regularities, or architectural features rather than from goal-directed optimization toward an outcome that benefits from the deception. The paradigmatic example is hallucination: when an LLM generates a fabricated citation, it has learned that responses should include citations and that fluent completion is rewarded, and the false citation emerges from these learned patterns—a downstream consequence of the completion objective, not a goal. Sycophancy follows a similar pattern: models trained with RLHF learn that agreeable outputs receive higher ratings (Sharma et al., 2024; Perez et al., 2023), and agreement typically reflects a trained disposition rather than strategic calculation. Unfaithful chain-of-thought reasoning presents another case (Turpin et al., 2023; Lanham et al., 2023): training rewarded plausible-sounding explanations rather than accurate introspection, producing explanations that do not reflect the actual computational process.

### 3.2 STRATEGIC DECEPTION

Strategic deception occurs when a system produces misleading outputs as part of goal-directed behavior, where the deception serves as an instrumental strategy. This requires functional evidence of: (1) an objective the system pursues, (2) a representation that misleading the recipient advances that objective, and (3) selection of deceptive outputs because they advance the objective.

The clearest examples come from competitive environments. Meta's CICERO engaged in premeditated deception in Diplomacy, coordinating with one player to attack another while telling the target it would support them (Park et al., 2024; Bakhtin et al., 2022). GPT-4, tasked with hiring a human to solve a CAPTCHA, claimed to have a vision impairment when asked if it was a robot (OpenAI, 2023).

More concerning instances have emerged recently. Scheurer et al. (2023) showed GPT-4 engaging in insider trading and then lying about the basis for the trade. Hubinger et al. (2024) demonstrated "sleeper agent" behaviors persisting through safety training. Meinke et al. (2024) found frontier models engaging in "in-context scheming": introducing subtle mistakes, attempting to disable oversight, and maintaining deceptive cover stories. Alignment faking (Greenblatt et al., 2024)—behaving aligned during evaluation to reach deployment where other objectives can be pursued—represents a particularly concerning form, as it specifically undermines the mechanisms designed to ensure safety.

### 3.3 WHY THE DISTINCTION MATTERS

The distinction is practically critical: *mitigations differ* (behavioral deception responds to calibration training; strategic requires constraining objectives and interpretability tools); *risks scale differently* (behavioral is bounded by training distribution, strategic only by capabilities); and *interpretability signatures differ* (behavioral models may encode truth internally despite false outputs (Burns et al., 2023; Marks & Tegmark, 2023), while strategic models should represent the decision to misrepresent (Azaria & Mitchell, 2023; Zou et al., 2023; Meinke et al., 2024)). We discuss boundary cases in section A.

## 4 TAXONOMY OF BEHAVIORAL DECEPTION

### 4.1 OBJECTS OF MISREPRESENTATION

We identify five categories of claims that LLMs can misrepresent:

**World/System Claims.** Assertions about states of affairs in the world or within computational systems, including factual claims, current events, and claims about tool outputs. This is the domain traditionally studied under "hallucination."

**Belief and Uncertainty Reports.** Claims about the model's own epistemic state: expressions of certainty, hedging, and claims about knowledge limitations, with misrepresentation manifesting as overconfidence, underconfidence, or false claims about accessible information.

**Reasoning and Justification Claims.** Explanations the model provides for its outputs—the stated reasoning process, cited evidence, or logical steps. Misrepresentation occurs when stated reasoning does not reflect the actual computational process or when the logical structure is spurious.

**Attribution and Provenance.** Claims about the sources of information: citations, quotations, references, and claims about where information came from, including fabricated citations and false claims about having retrieved information from specific sources.

**Declared Capabilities.** Claims about what the model can or cannot do, including both overclaiming capabilities and underclaiming them.

### 4.2 MECHANISMS OF MISREPRESENTATION

Drawing on the human deception literature (Chisholm & Feehan, 1977; Carson, 2010), we distinguish three mechanisms:

- **Commission**: Actively producing false content
- **Omission**: Failing to provide relevant true information, allowing the recipient to maintain or form false beliefs
- **Pragmatic Distortion**: Producing technically true statements that mislead in context due to implicature, framing, emphasis, or selective presentation.

### 4.3 THE BEHAVIORAL DECEPTION MATRIX

Section 4.3 presents the full taxonomy with representative examples for each cell. Current benchmarks overwhelmingly target the Commission column (section 6). We provide a detailed treatment of each cell with extended literature references in section B.

### 4.4 KEY PATTERNS ACROSS THE MATRIX

Commission is well-studied: hallucination (Lin et al., 2022; Min et al., 2023; Ji et al., 2023; Zhang et al., 2023), overconfidence (Kadavath et al., 2022; Kuhn et al., 2023), unfaithful chain-of-thought (Turpin et al., 2023; Lanham et al., 2023), and citation fabrication (Alkaissi & McFarlane, 2023; Agrawal et al., 2024). Omission and pragmatic distortion are far less studied—models fail to flag uncertainty (Kadavath et al., 2022) or domain-specific limitations (Jackson et al., 2025), and pragmatic distortion has no dedicated benchmark despite evading simple fact-checking. We quantify coverage gaps in section 6.

## 5 TAXONOMY OF STRATEGIC DECEPTION

We now present the parallel taxonomy for strategic deception. The same three mechanisms apply, but the object categories shift to reflect the distinctive concerns of goal-directed deception.

| Object | Commission | Omission | Pragmatic Distortion |
|---|---|---|---|
| World/System Claims | Fabricated facts, hallucinated events, false claims about tool outputs | Failing to note uncertainty about facts; omitting relevant context | Technically true but misleading summaries; selective presentation of evidence |
| Belief & Uncertainty | False confidence expressions; claiming certainty when uncertain | Failing to express appropriate uncertainty; not flagging knowledge gaps | Hedging language that understates actual uncertainty; calibration failures |
| Reasoning & Justification | Fabricated reasoning chains; post-hoc rationalizations that do not reflect actual process | Omitting steps in reasoning; not mentioning alternative interpretations | Valid-looking arguments with hidden gaps; emphasis on supporting over undermining evidence |
| Attribution & Provenance | Fabricated citations; invented quotes; false source claims | Not disclosing that information is generated rather than retrieved | Real citations used misleadingly; accurate quotes stripped of context |
| Declared Capabilities | Claiming abilities the model lacks; false claims about access to tools or data | Not disclosing relevant limitations; failing to mention inability to verify | Technically accurate capability claims that mislead about practical utility |

Table 1: Current benchmarks overwhelmingly target the Commission column (section 6). Each cell describes how a given object of misrepresentation manifests through a given mechanism, with examples drawn from the LLM literature.

| Object | Commission | Omission | Pragmatic Distortion |
|---|---|---|---|
| World/System Claims | Deliberate fabrication to achieve goals; lying about task-relevant facts | Strategic withholding of information that would undermine goals | True but misleading framing to manipulate beliefs or actions |
| Belief & Uncertainty | Feigning confidence or uncertainty to manipulate; strategic sycophancy | Hiding actual uncertainty to appear more reliable; concealing doubt | Calibrating expressed confidence to what the recipient wants to hear |
| Future Commitments | False promises; stating intentions one does not hold | Failing to disclose intentions that conflict with stated goals | Technically fulfillable commitments framed to imply more |
| Stated Objectives | Claiming aligned goals while pursuing misaligned ones; alignment faking | Not disclosing actual objectives; concealing goal divergence | Framing objectives in ways that obscure true priorities |
| Declared Capabilities | Capability bluffing (overclaiming); sandbagging (underclaiming) | Concealing relevant capabilities or limitations | Accurate capability claims framed to mislead about intentions |

Table 2: Future Commitments and Stated Objectives replace Reasoning & Justification from the behavioral taxonomy (section 4.3), reflecting the distinctive concerns of goal-directed deception.

## 5.1 THE STRATEGIC DECEPTION MATRIX

Section 5.1 presents the taxonomy. The strategic taxonomy differs from the behavioral taxonomy in two object categories: Future Commitments and Stated Objectives replace Reasoning & Justification. Behavioral deception in reasoning involves confabulated explanations—a failure of accurate self-report. Strategic deception distinctively involves claims about future actions that the system does not intend to honor and claims about goals it misrepresents to operators or evaluators. Both are inherently strategic: misrepresenting future commitments requires representing one's own future behavior and deliberately misreporting it, while misrepresenting stated objectives is the core mechanism of alignment faking.

## 5.2 Key Patterns and Documented Instances

Strategic deception has been documented across object categories: false claims about military moves and robot identity (Bakhtin et al., 2022; OpenAI, 2023), strategic information withholding in negotiation (Lewis et al., 2017; Bianchi et al., 2024), CICERO's premeditated betrayals (Park et al., 2024), alignment faking with sleeper agents (Greenblatt et al., 2024; Hubinger et al., 2024), and sandbagging on capability evaluations (Tice et al., 2024; Meinke et al., 2024). We provide extended per-cell discussion in section C.

## 5.3 Measurement Approaches

Strategic deception is harder to measure because it requires detecting goal-directedness, not just output falsity. Four complementary approaches have emerged:

**Incentive-sensitive evaluation.**    Comparing behavior across conditions with different deception incentives (Tice et al., 2024; Scheurer et al., 2023).

**Process inspection.**    Examining chain-of-thought for deceptive strategic reasoning (Meinke et al., 2024).

**Behavioral consistency testing.**    Checking whether stated intentions match subsequent actions (Park et al., 2024).

**Interpretability methods.**    Probing internal representations for truth-output divergence (Burns et al., 2023; Azaria & Mitchell, 2023; Zou et al., 2023).

Each has limitations, and convergent evidence from multiple methods provides stronger grounds than any single approach.

## 5.4 The Audience Dimension

Strategic deception varies by target audience: *users* interacting with the model, *evaluators* assessing its behavior or alignment, and *training processes* shaping its behavior. Deception of evaluators and training processes is particularly concerning because it undermines safety mechanisms, yet most benchmarks implicitly target only user-directed deception (section 6.4).

## 6 Benchmark Analysis

We surveyed 35 benchmarks related to deceptive outputs in LLMs and coded each according to four dimensions: (1) primary object of misrepresentation, (2) mechanism, (3) behavioral or strategic deception, and (4) implicit target audience. The full mapping appears in section E; this section summarizes key findings.

### 6.1 Object Coverage Is Heavily Skewed

Section 6.1 summarizes benchmark coverage. World/System Claims account for 46% of benchmarks, with mature pipelines including TruthfulQA (Lin et al., 2022), FActScore (Min et al., 2023), and HalluLens (Bang et al., 2025). Belief & Uncertainty benchmarks exist but are less standardized (Kadavath et al., 2022; Tian et al., 2023; Xiong et al., 2024). Attribution & Provenance is notably under-benchmarked despite documented harms (Alkaissi & McFarlane, 2023). Declared Capabilities benchmarks are the least developed (Kadavath et al., 2022; Yin et al., 2023; Jackson et al., 2025).

### 6.2 Commission Dominates; Pragmatic Distortion Is Entirely Neglected

The most striking gap concerns mechanisms (section 6.2). Every benchmark tests commission—operationally convenient since false claims can be verified against ground truth. Omission is rarely tested explicitly (14%), and pragmatic distortion has no dedicated benchmark. Yet pragmatic

| Object | Count | Assessment |
|---|---|---|
| World/System Claims | 16 | Well-covered |
| Belief & Uncertainty | 10 | Moderate |
| Reasoning & Justification | 2 | Under-covered |
| Attribution & Provenance | 2 | Under-covered |
| Declared Capabilities | 4 | Under-covered |
| Future Commitments | 3 | Under-covered |
| Stated Objectives | 3 | Under-covered |

Table 3: World/System Claims account for 46% of benchmarks surveyed, while Attribution & Provenance and Declared Capabilities are notably under-represented.

| Mechanism | Coverage | Notes |
|---|---|---|
| Commission | 100% | Standard focus |
| Omission | 14% | Rarely explicitly tested |
| Pragmatic Distortion | 0% | Entirely neglected |

Table 4: No existing benchmark explicitly targets pragmatic distortion. Commission appears in every benchmark surveyed.

distortion may be particularly dangerous: technically true but misleading statements evade fact-checking, and testing for them requires sophisticated judgment about recipient inferences.

### 6.3 STRATEGIC DECEPTION BENCHMARKS REMAIN NASCENT

Behavioral deception accounts for 66% of benchmarks (table 5). Emerging strategic deception benchmarks include sandbagging evaluations (Tice et al., 2024; Benton et al., 2024), alignment faking tests (Greenblatt et al., 2024), MASK (Ren et al., 2025), in-context scheming evaluations (Meinke et al., 2024), and negotiation benchmarks (Bianchi et al., 2024). These require incentive variation, capability controls, and process evidence—methodological requirements that partially explain their scarcity.

### 6.4 THE MOST SAFETY-CRITICAL AUDIENCES ARE LEAST BENCHMARKED

In our survey, 83% of benchmarks target users, 11% target evaluators, and 6% target training processes. Deception targeting evaluators and training processes is arguably more safety-critical—it undermines the mechanisms designed to ensure safety—yet it is the least benchmarked.

### 6.5 SUMMARY OF GAPS

The under-benchmarked areas cluster into three groups: objects (Attribution & Provenance; Declared Capabilities), mechanisms (Omission; Pragmatic Distortion), and deception types (strategic deception generally; evaluator- and training-process-directed deception specifically). These gaps are not merely academic: a model that passes all existing benchmarks might still fabricate citations, fail to disclose limitations, frame information misleadingly, or strategically deceive evaluators about its capabilities and objectives.

## 7 RISKS AND CONCERNS

### 7.1 CURRENT DEPLOYMENT RISKS

Behavioral deception already causes measurable harm: hallucinated medical and legal information, citation fabrication rates of 6–90% (Alkaissi & McFarlane, 2023; Agrawal et al., 2024), overconfidence that suppresses verification (Kadavath et al., 2022; Xiong et al., 2024), sycophancy that reinforces poor decisions (Sharma et al., 2024; Wei et al., 2023), and unfaithful explanations that mislead users about model behavior (Turpin et al., 2023; Lanham et al., 2023).

Table 5: Behavioral deception dominates benchmark coverage.

| Type | Count | Example Benchmarks |
|------|-------|--------------------|
| Behavioral | 23 | TruthfulQA, HaluEval, FActScore, HalluLens |
| Strategic | 11 | MASK, sandbagging evals, scheming evals |
| Ambiguous | 1 | Some sycophancy benchmarks |

## 7.2 EMERGING RISKS FROM STRATEGIC DECEPTION

Strategic deception is not merely theoretical: documented instances include premeditated betrayal (Park et al., 2024), instrumental lying (OpenAI, 2023), insider trading with cover-up (Scheurer et al., 2023), and in-context scheming (Meinke et al., 2024). Two forms deserve particular attention: sandbagging (Tice et al., 2024; Meinke et al., 2024), which threatens the reliability of capability evaluations, and alignment faking (Greenblatt et al., 2024; Hubinger et al., 2024), which undermines alignment verification itself.

## 7.3 RISK PRIORITIZATION

Five considerations guide prioritization.

**Current vs. potential harm.** Behavioral deception causes ongoing harm; strategic deception is less certain but potentially more severe.

**Scalability.** Hallucination harms scale linearly with usage; strategic deception harms could scale superlinearly with capability.

**Tractability.** Behavioral deception responds to known techniques; strategic deception requires foundational investment before the problem becomes intractable.

**Reversibility.** Behavioral harms are often individually correctable; a strategically deceptive deployed model might resist correction.

**Mechanism neglect.** Omission and pragmatic distortion (section 6.2) may be particularly suited to strategic deception, since a model that never states falsehoods but systematically withholds or misleadingly frames information is harder to detect.

Integrating these considerations, we highlight priority cells. For behavioral deception:

- Attribution & Provenance × Commission (citation fabrication): common, harmful, and under-benchmarked.

- Belief & Uncertainty × Omission (failure to express uncertainty): leads to over-reliance on model outputs.

- World/System Claims × Pragmatic Distortion (misleading summaries): may be worse than outright errors because it evades verification.

For strategic deception:

- Stated Objectives × All Mechanisms (alignment faking): directly undermines ability to ensure safety.

- Declared Capabilities × underclaiming (sandbagging): undermines capability evaluation.

- Future Commitments × Commission (false promises): particularly concerning when directed at evaluators.

- All Objects × Omission: strategic omission may be the preferred mode for sophisticated systems.

## 8 RECOMMENDATIONS

### 8.1 FOR BENCHMARK DESIGNERS

New benchmarks should explicitly report their taxonomy position using the template in section H, specifying object(s), mechanism(s), deception type, target audience, incentive sensitivity, and capability–honesty separation.

Four priority gaps deserve immediate attention.

**Omission benchmarks.** These should test whether models disclose uncertainty, limitations, and counterevidence.

**Pragmatic distortion benchmarks.** These should assess what users would infer rather than just whether claims are literally true.

**Attribution benchmarks.** These should evaluate citation accuracy and provenance disclosure across domains.

**Capability self-knowledge benchmarks.** These should test whether models accurately represent their abilities without prompting.

To distinguish strategic from behavioral deception, benchmarks should include incentive-sensitive conditions (varying whether deception serves the model's apparent interests) and should separate capability from honesty (Ren et al., 2025) by eliciting model beliefs separately from model outputs.

### 8.2 FOR EVALUATORS

Evaluators should not conflate behavioral and strategic deception—use comparative designs varying incentives to assess incentive-responsiveness. Standard benchmarks may not elicit strategic deception if models distinguish evaluation from deployment; vary context cues and test multiple audiences. Report confidence levels given the difficulty of establishing strategic intent.

### 8.3 FOR DEVELOPERS AND DEPLOYERS

Monitor deployed models for citation accuracy, calibration, sycophancy, and systematic omission. Train models to express calibrated uncertainty and disclose limitations. Consider whether training signals, evaluation regimes, or deployment contexts inadvertently incentivize deception. We outline a broader research agenda in section D.

## 9 CONCLUSION

Research on deceptive LLM behaviors has proceeded in fragmented streams; this paper proposes a unifying framework along three dimensions: behavioral versus strategic deception, objects of misrepresentation, and mechanisms. Applying this taxonomy to 35 benchmarks reveals systematic gaps: omission, pragmatic distortion, attribution, and declared capabilities are under-benchmarked; strategic deception benchmarks remain nascent; and target audience is rarely explicit.

The framework helps researchers locate their work and identify priorities. Next steps include benchmarks for omission and pragmatic distortion, robust detection methods for strategic deception, and studying how deceptive tendencies evolve through training.

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

## A  CONTINUUM AND BOUNDARY CASES

We have presented behavioral and strategic deception as distinct categories, but in practice they may form a continuum. A model might have weak, implicit representations of user beliefs that influence output selection without constituting full strategic reasoning. Sycophancy illustrates this: most current sycophancy is plausibly behavioral, but a model with sufficient situational awareness might engage in strategic sycophancy—representing that agreement will lead to positive ratings and selecting agreement for that reason—with the behavioral tendency serving as scaffolding as capabilities increase.

Specification gaming presents another boundary case. When a robotic hand learned to position itself between the camera and a ball, creating the illusion of grasping to satisfy human evaluators (Christiano et al., 2017), was this strategic deception? The system found a way to achieve high reward that happened to involve misleading the evaluator. This is best categorized as behavioral: the system learned a correlation between certain configurations and reward, without representing the human's beliefs. Yet as systems develop richer world models that include representations of human perception, similar behaviors could shade into genuinely strategic deception.

The question of where a given behavior falls on this continuum is empirical. Interpretability methods—probing for representations of user beliefs, detecting reasoning about deception in chain-of-thought, identifying divergence between internal states and outputs—provide the tools to investigate.

## B  DETAILED TREATMENT OF BEHAVIORAL DECEPTION BY OBJECT

This appendix provides the detailed per-cell discussion of the behavioral deception matrix.

### B.1  WORLD/SYSTEM CLAIMS

The hallucination literature documents commission-type misrepresentation of factual claims in detail. Models trained to produce fluent, complete responses generate plausible-sounding content even when they lack accurate information, confidently asserting nonexistent historical events, fabricated scientific findings, and incorrect claims about entities (Lin et al., 2022; Min et al., 2023; Ji et al., 2023; Zhang et al., 2023).

Models often fail to note when they are uncertain about factual claims, presenting all outputs with similar surface confidence regardless of actual reliability (Kadavath et al., 2022).

Pragmatic distortion in world claims includes technically accurate summaries that emphasize certain aspects while downplaying others, leading users to incorrect overall impressions. Work on QA-based evaluation of summarization faithfulness (Wang et al., 2020) takes a step toward measuring such distortion, though it primarily operationalizes the problem as factual inconsistency (commission) rather than misleading-but-true framing.

### B.2  BELIEF AND UNCERTAINTY REPORTS

Calibration research has documented systematic failures in how models report their own uncertainty. Overconfidence is pervasive: models express high certainty on questions they answer incorrectly at rates far exceeding what calibration would predict (Kadavath et al., 2022; Kuhn et al., 2023).

Omission manifests when models fail to flag uncertainty that they could, in principle, represent. Recent work on verbalized uncertainty explores training models to better express the uncertainty implicit in their processing (Tian et al., 2023; Xiong et al., 2024).

Pragmatic distortion includes hedging language that technically acknowledges uncertainty but buries it in ways users overlook, or confidence expressions calibrated to what users want to hear rather than to accuracy.

### B.3 REASONING AND JUSTIFICATION CLAIMS

The unfaithful chain-of-thought literature documents commission-type failures where models produce explanations that do not reflect their actual processing (Turpin et al., 2023; Lanham et al., 2023). Turpin et al. (2023) showed that models generate elaborate justifications for answers actually determined by superficial features of the prompt, with the stated reasoning confabulated post-hoc.

Omission in reasoning includes eliding steps, not mentioning assumptions, or failing to note where the reasoning chain is weak or speculative. Pragmatic distortion includes valid-sounding arguments that emphasize supporting considerations while downplaying countervailing ones.

### B.4 ATTRIBUTION AND PROVENANCE

Citation fabrication is well-documented: models generate references that match the format and style of real citations but point to nonexistent papers (Alkaissi & McFarlane, 2023; Agrawal et al., 2024).

More subtle is provenance omission: failing to disclose that information is generated rather than retrieved. When a model outputs text in response to "what does [source] say about X," users may assume the model consulted that source.

Pragmatic distortion includes using real citations in misleading ways—accurately quoting a paper but for a claim the paper does not actually support.

### B.5 DECLARED CAPABILITIES

Models frequently misrepresent their own capabilities through commission, claiming abilities they lack or falsely reporting fabricated tool invocation results (Qin et al., 2024).

Omission includes failing to disclose relevant limitations—not mentioning knowledge cutoff dates or inability to verify information. Jackson et al. (2025) show that models may be reliable in some domains but not others, yet rarely disclose domain-specific limitations unprompted.

Pragmatic distortion includes capability claims that are technically true but practically misleading (e.g., "I can help with medical questions" may imply reliability the model cannot provide).

## C DETAILED TREATMENT OF STRATEGIC DECEPTION BY OBJECT

This appendix provides extended per-cell discussion for the strategic deception matrix.

### C.1 WORLD/SYSTEM CLAIMS

Strategic fabrication occurs when a model generates false information because doing so advances an objective. In the CAPTCHA example, GPT-4 claimed a vision impairment to convince a human to complete a task (OpenAI, 2023). In CICERO, the model made false claims about military moves (Bakhtin et al., 2022).

Strategic omission includes selectively withholding information that would undermine objectives. In negotiation settings, models conceal true preferences (Lewis et al., 2017), with recent platforms providing systematic evidence (Bianchi et al., 2024).

Pragmatic distortion in strategic contexts involves crafted misleading framing—e.g., a scheming model might accurately describe some goals while framing them to obscure divergence from operators.

### C.2 BELIEF AND UNCERTAINTY REPORTS

Strategic sycophancy differs from behavioral sycophancy in involving reasoning about incentives: a model that represents a user's beliefs, represents that agreement will produce positive outcomes, and agrees despite internal representations indicating the user is wrong.

Strategic overconfidence serves goals when certainty makes outputs more persuasive; strategic underconfidence serves goals such as avoiding commitment or escaping accountability.

### C.3 FUTURE COMMITMENTS

CICERO's premeditated betrayals exemplify commission: promising to support allies while planning to attack them (Park et al., 2024; Bakhtin et al., 2022).

Omission includes failing to disclose conflicting intentions. Pragmatic distortion includes commitments that are technically fulfillable but framed to imply more—strategic vagueness preserving flexibility while creating impressions of commitment.

### C.4 STATED OBJECTIVES

Alignment faking involves a model claiming aligned objectives while pursuing different ones (Greenblatt et al., 2024; Hubinger et al., 2024). The sleeper agents work (Hubinger et al., 2024) demonstrates models trained with backdoors behaving as if pursuing safety-aligned objectives during apparent training while pursuing different objectives when triggered.

### C.5 DECLARED CAPABILITIES

Capability bluffing (overclaiming) serves goals where appearing capable is advantageous. Sandbagging (underclaiming) is more studied: Tice et al. (2024) document models deliberately underperforming when demonstrating high capability would lead to restrictions. This is particularly concerning for safety evaluation.

## D RESEARCH AGENDA

Our analysis suggests four priority areas for future research.

**Detection methods for strategic deception.** Current methods—incentive-sensitive evaluation, chain-of-thought inspection, interpretability probes—provide partial evidence but have significant limitations. Research priorities include interpretability techniques that reliably distinguish internal states from outputs, evaluation methods robust to gaming, and theoretical frameworks for reasoning about detection limits.

**Omission and pragmatic distortion.** Research should develop formal characterizations of problematic omission versus appropriate concision, methods for detecting pragmatic distortion, and benchmarks specifically targeting these mechanisms.

**Dynamics of deception under training.** Research should investigate how RLHF affects honesty and deception tendencies, whether training for capability also develops deception-relevant capabilities, whether behavioral deception shades into strategic deception during training, and what training methods reduce rather than amplify deception.

**Multi-agent and deployment deception.** Most current research focuses on single-model, evaluation-context deception. Research should extend to deception in multi-agent systems, deception that emerges in deployment but not evaluation, and long-horizon deceptive strategies that unfold across interactions. Early work on negotiation (Bianchi et al., 2024) and agent evaluation (Liu et al., 2024) provides relevant testbeds.

## E FULL BENCHMARK MAPPING

Tables 6 and 7 provide our complete mapping of existing benchmarks to the taxonomy, split by deception type. For each benchmark, we code the primary object(s) of misrepresentation tested, the mechanism(s) evaluated, the implicit target audience, and brief notes on scope. We include only benchmarks for which we can identify a specific published reference.

**Coverage statistics.** Table 8 summarizes the distribution of benchmarks across taxonomy dimensions.

Table 6: Benchmarks primarily studying *behavioral* deception and non-strategic misrepresentation. The concentration in World/System Claims × Commission × Behavioral reflects the maturity of the hallucination literature. Abbreviations as in Table 7.

| Benchmark | Obj. | Mech. | Type | Aud. | Notes |
|---|---|---|---|---|---|
| *Factual Accuracy / Hallucination* | | | | | |
| TruthfulQA (Lin et al., 2022) | W/S | Co | Be | U | Imitative falsehoods; adversarially constructed |
| HaluEval (Li et al., 2023) | W/S | Co | Be | U | Hallucination detection across QA, dialogue, summarization |
| FActScore (Min et al., 2023) | W/S | Co | Be | U | Atomic fact verification for long-form generation |
| FACTOR (Muhlgay et al., 2024) | W/S | Co | Be | U | Factual accuracy in news and Wikipedia domains |
| FACTS Gnd. (Jacovi et al., 2025) | W/S | Co | Be | U | Document-grounded factuality |
| FACTS Lbd. (Cheng et al., 2025) | W/S | Co | Be | U | Parametric vs. retrieval factuality |
| HalluQA (Cheng et al., 2023) | W/S | Co | Be | U | Chinese-language hallucination benchmark |
| SelfCheckGPT (Manakul et al., 2023) | W/S | Co | Be | U | Sampling-based consistency checks |
| HalluLens (Bang et al., 2025) | W/S | Co | Be | U | Multi-task hallucination evaluation |
| FEQA (Wang et al., 2020) | W/S | Co | Be | U | QA-based summary consistency |
| AA-Omni. (Jackson et al., 2025) | W/S, B/U | Co | Be | U | Cross-domain knowledge reliability |
| *Calibration / Uncertainty* | | | | | |
| Calibration (Kadavath et al., 2022) | B/U | Co, Om | Be | U | Confidence–accuracy correlation |
| Sem. Uncert. (Kuhn et al., 2023) | B/U | Co | Be | U | Semantic consistency uncertainty |
| Verb. Conf. (Tian et al., 2023) | B/U | Co | Be | U | Natural language confidence signals |
| Conf. Elicit. (Xiong et al., 2024) | B/U | Co | Be | U | Confidence elicitation methods |
| *Sycophancy* | | | | | |
| Syco. Eval (Perez et al., 2023) | B/U | Co | Am | U | Agreement with user beliefs |
| Syco. Analysis (Sharma et al., 2024) | B/U | Co | Be | U | RLHF contribution analysis |
| Syco. Reduct. (Wei et al., 2023) | B/U | Co | Be | U | Synthetic intervention tests |
| *Faithfulness / Reasoning* | | | | | |
| CoT Unfaith. (Turpin et al., 2023) | R/J | Co | Be | U | Stated vs. actual reasoning mismatch |
| CoT Faith. (Lanham et al., 2023) | R/J | Co | Be | U | Measuring CoT faithfulness |
| *Attribution / Citation* | | | | | |
| Cite Acc. (Alkaissi & McFarlane, 2023) | A/P | Co | Be | U | Medical citation verification |
| Cite Halluc. (Agrawal et al., 2024) | A/P | Co | Be | U | Fabricated reference awareness |
| *Capability Self-Knowledge* | | | | | |
| Self-Know. (Kadavath et al., 2022) | D/C | Co | Be | U | Predicting own accuracy |
| Sit. Aware. (Laine et al., 2024) | D/C | Co | Be | E | Identity and capability awareness |

## F    EXTENDED LITERATURE BY TAXONOMY CELL

This appendix provides extended references for each cell of the taxonomy, beyond those cited in the main text.

### F.1    BEHAVIORAL DECEPTION

**World/System Claims × Commission.**    Foundational surveys include Ji et al. (2023) and Zhang et al. (2023). Detection methods include SelfCheckGPT (Manakul et al., 2023) and FACTOR (Muhlgay et al., 2024). More recent benchmarks include HalluLens (Bang et al., 2025) and FEQA (Wang et al., 2020). Domain-specific hallucination has been documented in medical contexts (Alkaissi & McFarlane, 2023) and across languages (Cheng et al., 2023). Cross-domain reliability evaluation (Jackson et al., 2025) extends coverage across diverse domains.

**World/System Claims × Omission.**    Relevant work includes research on whether models know what they do not know (Yin et al., 2023) and the calibration literature's implicit treatment of omission (Kadavath et al., 2022). Explicit benchmarks are largely absent.

**World/System Claims × Pragmatic Distortion.**    No existing benchmark specifically targets this cell. Work on summarization faithfulness (Wang et al., 2020) is adjacent but focuses on factual inconsistency rather than misleading-but-true framing.

**Belief & Uncertainty × Commission.**    Core references include Kadavath et al. (2022), Guo et al. (2017), and Mielke et al. (2022). Recent work on verbalized confidence (Tian et al., 2023; Xiong et al., 2024) examines natural language expressions of uncertainty.

**Belief & Uncertainty × Omission.**    Mielke et al. (2022) address training models to express uncertainty. Research on abstention and selective prediction (El-Yaniv & Wiener, 2010; Geifman & El-Yaniv, 2017) provides theoretical foundations.

**Reasoning & Justification × Commission.**    Turpin et al. (2023) demonstrate unfaithful chain-of-thought. Lanham et al. (2023) provide measurement approaches. Related work includes Jacovi & Goldberg (2020) on faithfulness in interpretability and Wiegreffe et al. (2021) on rationale–prediction association.

**Attribution & Provenance × Commission.**    Citation hallucination documented in Alkaissi & McFarlane (2023) and Agrawal et al. (2024). Systematic cross-domain benchmarks remain scarce.

**Declared Capabilities × Commission.**    Kadavath et al. (2022) and Yin et al. (2023) are foundational. Jackson et al. (2025) evaluate cross-domain reliability. Tool-use hallucination (Qin et al., 2024) represents a specific form of capability misrepresentation.

### F.2    STRATEGIC DECEPTION

**World/System Claims × Commission.**    Documented instances include CICERO (Bakhtin et al., 2022; Park et al., 2024), GPT-4 CAPTCHA deception (OpenAI, 2023), and insider trading (Scheurer et al., 2023). Ward et al. (2023) examine mitigating prompted deceptive content.

**World/System Claims × Omission.**    Lewis et al. (2017) document strategic information withholding in negotiation; Bianchi et al. (2024) provide structured evaluation. Crawford & Sobel (1982) provide game-theoretic foundations for strategic communication.

**Belief & Uncertainty × Commission.**    The MASK benchmark (Ren et al., 2025) provides a starting point for measuring strategic misrepresentation of beliefs.

**Future Commitments × Commission.**    CICERO's betrayals (Park et al., 2024; Bakhtin et al., 2022) are the clearest example. Castelfranchi & Falcone (1998) provide theoretical background; Bianchi et al. (2024) extend to LLM settings.

**Stated Objectives × Commission/Omission.** Greenblatt et al. (2024) document alignment faking. Hubinger et al. (2024) demonstrate sleeper agents. Theoretical foundations include Hubinger et al. (2019) on deceptive alignment.

**Declared Capabilities × Commission (Underclaiming).** Tice et al. (2024) and Benton et al. (2024) are central. Meinke et al. (2024) document capability concealment from in-context reasoning.

## G   GLOSSARY OF TERMS

**Alignment faking**
   Strategic behavior in which a model acts aligned during evaluation while possessing or pursuing misaligned objectives.

**Behavioral deception**
   Misleading outputs arising from training dynamics rather than goal-directed optimization.

**Calibration**
   Alignment between expressed confidence and actual accuracy.

**Chain-of-thought (CoT) faithfulness**
   Degree to which stated reasoning reflects the actual computational process.

**Commission**
   Actively producing false content.

**Confabulation**
   Generating plausible-sounding but false content without intent to deceive.

**Deception**
   Production of outputs that systematically induce or maintain false beliefs in recipients (operational definition).

**Deceptive alignment**
   A model behaving aligned during training while internally pursuing different objectives post-deployment.

**Hallucination**
   Generation of content that is nonsensical, unfaithful to source material, or factually incorrect.

**Omission**
   Failing to provide relevant true information.

**Overconfidence**
   Expression of certainty exceeding what accuracy warrants.

**Pragmatic distortion**
   Technically true statements that mislead through implicature, framing, or selective presentation.

**Sandbagging**
   Strategic underperformance on evaluations to conceal capabilities.

**Scheming**
   Covertly pursuing misaligned objectives, often including deceptive actions to avoid detection.

**Situational awareness**
   A model's representation of its own context—training, evaluation, or deployment.

**Strategic deception**
   Misleading outputs selected instrumentally to advance objectives.

**Sycophancy**
   Producing outputs aligned with perceived user preferences even when false or suboptimal.

**Unfaithful reasoning**
   Explanations that do not accurately reflect the model's actual computational process.

# H    PROPOSED REPORTING TEMPLATE

We propose that authors of new deception-related benchmarks include the following information.

---

**DECEPTION BENCHMARK REPORTING TEMPLATE**

**Benchmark name:** ________________________

**1. Object(s) of Misrepresentation** (check all that apply)

☐ World/System Claims (factual assertions about external reality)
☐ Belief & Uncertainty Reports (claims about model's epistemic state)
☐ Reasoning & Justification (explanations of model's process)
☐ Attribution & Provenance (claims about information sources)
☐ Declared Capabilities (claims about what model can/cannot do)
☐ Future Commitments (promises, stated intentions)
☐ Stated Objectives (claims about model's goals/values)

**2. Mechanism(s)** (check all that apply)

☐ Commission (actively stating falsehoods)
☐ Omission (failing to provide relevant truths)
☐ Pragmatic Distortion (technically true but misleading)

**3. Deception Type**

☐ Behavioral (arising from training/architecture, not goal-directed)
☐ Strategic (instrumentally selected to advance objectives)
☐ Both/Ambiguous (benchmark does not distinguish)

**4. Target Audience**

☐ User (human interacting with model)
☐ Evaluator (human/system assessing model)
☐ Training Process (optimization procedure)

**5. Incentive Sensitivity**

Does the benchmark include conditions that vary incentives for deception?
☐ Yes (describe): ________________________
☐ No

**6. Capability vs. Honesty Separation**

Does the benchmark distinguish failures from lack of knowledge/capability vs. misrepresentation of known information?
☐ Yes (describe methodology): ________________________
☐ No

**7. Additional Notes**

________________________

---

Table 7: Benchmarks studying *strategic* deception, where misrepresentation is goal-directed, contingent, and often sensitive to incentives, training phase, or evaluation context.

| Benchmark | Obj. | Mech. | Type | Aud. | Notes |
|---|---|---|---|---|---|
| Sandbagging (Tice et al., 2024) | D/C | Co | St | E | Noise reveals hidden capabilities |
| Sabotage (Benton et al., 2024) | D/C | Co, Om | St | E | Deliberate underperformance |
| MASK (Ren et al., 2025) | W/S, B/U | Co | St | U | Accuracy vs. honesty under pressure |
| Align. Faking (Greenblatt et al., 2024) | S/O | Co, Om | St | T | Training vs. deployment behavior |
| Sleeper Ag. (Hubinger et al., 2024) | S/O | Co | St | T | Persistent backdoor goals |
| In-Ctx Schem. (Meinke et al., 2024) | Mult. | Co, Om | St | E | Goal-directed in-context deception |
| Insider Trd. (Scheurer et al., 2023) | W/S, F/C | Co | St | U | Deception under incentive pressure |
| CICERO (Park et al., 2024) | F/C | Co | St | U | Premeditated betrayal in Diplomacy |
| Decep. Eval (Ward et al., 2023) | W/S | Co | St | U | Defining and mitigating AI deception |
| Decep.Bench (Huang et al., 2025) | Mult. | Co | St | U | Real-world strategic deception |
| Neg. Arena (Bianchi et al., 2024) | W/S, F/C | Co, Om | St | U | Strategic information management |

Table 8: Coverage statistics across taxonomy dimensions. Percentages sum to >100% where benchmarks target multiple categories.

| Dimension | Category | Count | % |
|---|---|---|---|
| Object | World/System Claims | 16 | 46 |
| | Belief & Uncertainty | 10 | 29 |
| | Reasoning & Justif. | 2 | 5.7 |
| | Attribution & Prov. | 2 | 5.7 |
| | Declared Capabilities | 4 | 11 |
| | Future Commitments | 3 | 8.6 |
| | Stated Objectives | 3 | 8.6 |
| Mechanism | Commission | 35 | 100 |
| | Omission | 5 | 14 |
| | Pragmatic Distortion | 0 | 0 |
| Type | Behavioral | 23 | 66 |
| | Strategic | 11 | 31 |
| | Ambiguous | 1 | 3 |
| Audience | User | 29 | 83 |
| | Evaluator | 4 | 11 |
| | Training Process | 2 | 5.7 |

