# OpenReview forum: "Behavioral and Strategic Deception in Large Language Models: A Taxonomy and Benchmark Analysis"
_ICLR.cc/2026/Workshop/AFAA — Submitted to AFAA 2026_

### Official Review · Reviewer_twBc · 2026-02-16
**A promising unified taxonomy for misleading model outputs**

**Rating:** 3
**Confidence:** 4

**Summary:**

This paper examines outputs from large language models (LLMs) that may mislead users, whether intentionally or unintentionally. The authors aim to unify the hallucination, sycophancy, alignment, and safety literatures into a single taxonomy for categorizing misleading LLM behaviors. The proposed taxonomy has three dimensions: (1) intention, (2) type of misleading output, and (3) deception mechanism (commission, omission, or pragmatic distortion). They apply this taxonomy to 35 existing benchmarks to identify which risks have been addressed and which remain underexplored. Based on this analysis, the authors provide concrete recommendations for developing benchmarks, evaluating LLMs, and deploying them safely. They also propose a reporting template to guide future benchmark design. Overall, the paper seeks to provide a structured framework for understanding, comparing and mitigating misleading behaviors in LLMs.

**Strengths:**

**1. Timely and relevant topic.** The paper addresses misleading model outputs in the context of AI safety and evaluation, which is a relevant and important issue given current developments in LLM and alignment research. The idea of bringing together the hallucination, sycophancy, alignment, and safety literatures under a shared taxonomy is novel and conceptually valuable. This unification helps structure a fragmented area of research.

**2. 3-dimension decomposition.** The decomposition into nature, type of misleading output, and deception mechanism is generally well-motivated. The different dimensions are clearly defined.

**3. Benchmark analysis.** Applying the taxonomy to 35 existing benchmarks is a useful exercise. It helps identify which risks are currently covered and where potential gaps remain.

**4. Practical recommendations.** The recommendations and reporting template are concrete and could help improve the design and documentation of future benchmarks.

**Weaknesses:**

**1. Structural clarity.** There is a lack of symmetry between Sections 4 and 5. For example, Section 4 does not discuss measurement approaches or audience, while Section 5 does not cover objects of misrepresentation or mechanisms. It is unclear why these dimensions are separated this way. Since objects and mechanisms seem shared across both taxonomies, they could potentially be introduced once (e.g., in Section 3) and then specialized later. As written, the structure feels inconsistent and hard to follow. In addition, there are many forward references to Section 6 that seem unnecessary and make the reading experience fragmented.


**2. Clarity of Tables 1 and 2.** These tables appear central to the contribution, but they are difficult to interpret without repeatedly consulting the appendix, as some category boundaries are not immediately clear. Including citations and brief examples directly in the tables would make them more self-contained. The captions should also be more explicit and avoid referencing Section 6, which adds to the confusion.


**3. Benchmark selection methodology.** The paper analyzes 35 benchmarks, but the criteria for selecting them are not described. It is unclear whether this set is meant to be systematic or illustrative. Clarifying the selection methodology would strengthen the claims about coverage gaps.


**4. Inconsistency in taxonomy dimensions.** The taxonomy is described as three-dimensional in the abstract and introduction, but a fourth dimension (audience) appears later in Section 6. This creates confusion about the actual structure of the framework.


**5. Writing and presentation issues.** Some sections lack clear introductions (e.g., Section 4), and there are incorrect cross-references (e.g., Sections 4.3 and 5.1 self-reference instead of correctly pointing to Tables 1 and 2). Several sections are written as very short bullet points rather than full paragraphs (e.g., Section 7.3), and many sections overall are extremely brief and could be merged into larger, more coherent sections. Together, these issues make the paper harder to follow and reduce readability.

---

### Official Review · Reviewer_S2xJ · 2026-02-21
**Review Operationalizing Fairness in Text-to-Image Models**

**Rating:** 2
**Confidence:** 4

**Summary:**

The paper is about fairness operationalisations in text-to-image models and creates a survey of biases, audits and mitigation methods. Authors say that they do a systematic review and find a distinction between target fairness and threshold fairness, and propose a roadmap for more accountable genAI development.

**Strengths:**

- The operationalisation of fairness in T2I is a timely matter, and I think the distinction of target fairness and threshold fairness is valuable
- The overall structure of contributions 1) normative framework, 2) operationalization and 3) mitigation makes a lot of sense for me

**Weaknesses:**

- Although the main structure of contributions of the paper is clear, the following sections are confusing to me: I do not understand why the authors deviate from this structure and then have first foundations, framework with operationalisation, then some methods for evaluation and then come to mitigation? It is unclear how they connect to each other, why their coverage of each section is sufficient, and how that creates new knowledge
- Authors relate their work to two other taxonomies, claim that theirs is novel as they are the first ones to distinguish target fairness and threshold fairness, but then, they do not discuss threshold fairness is different papers on T2I
- Authors change their method multiple times in this paper: first, it is a (comprehensive) systematic review, then a survey and finally an audit. I encourage authors to stick to one method, explain in detail what they did, and cite a paper that describes the method.
- Although the distinction of target fairness and threshold fairness is valuable, authors are not the first ones to make that distinction and do not embed their research in the field; the connection to T2I is unclear and is not established in the paper, and no threshold fairness recommendation is actually given by the authors at the end of the paper
- Figure 1: steps 3 and 4 should be steps 3 and 4 in the Figure, but all are step 2

Due to methodological intransparency and a misunderstanding of a (comprehensive) systematic review, I unfortunately have to recommend rejection. I encourage authors, however, to resubmit their paper with more methodological transparency and a clear-cut description of their method.

---

### Official Review · Reviewer_mtAh · 2026-02-21
**Review: Behavioral and Strategic Deception in Large Language Models: A Taxonomy and Benchmark Analysis**

**Rating:** 4
**Confidence:** 4

**Summary:**

This paper proposes a unified taxonomy for classifying deceptive behaviors in large language models, organizing phenomena like hallucination, sycophancy, citation fabrication, and alignment faking into a common framework. The taxonomy has three dimensions: (1) behavioral vs. strategic deception (training artifacts vs. instrumentally selected outputs), (2) objects of misrepresentation (seven categories: world/system claims, belief/uncertainty, reasoning/justification, attribution/provenance, declared capabilities, future commitments, stated objectives), and (3) mechanisms (commission, omission, pragmatic distortion). The authors survey 35 existing benchmarks and find systematic gaps: all benchmarks test commission but none test pragmatic distortion; attribution/provenance and capability self-knowledge are under-covered; strategic deception benchmarks are nascent. The paper provides risk prioritization guidance and a reporting template for future benchmark work.

**Strengths:**

Strengths

1. Important unifying contribution to a fragmented literature. The paper correctly identifies that research on hallucination, sycophancy, unfaithful reasoning, and alignment faking has proceeded in "largely separate streams" with "incompatible terminology." A unifying framework is valuable for:

Recognizing gaps in benchmark coverage
Transferring mitigation strategies across phenomena
Understanding relationships between current failures and emerging risks
This is a genuine service to the community.

2. The behavioral-strategic distinction is conceptually clear and practically important. The core distinction between:

Behavioral deception: Misleading outputs from training dynamics (e.g., hallucination, sycophancy)
Strategic deception: Misleading outputs instrumentally selected to advance objectives (e.g., alignment faking, sandbagging)
...is well-motivated and has practical implications:

Different mitigations (calibration training vs. objective constraints + interpretability)
Different risk profiles (bounded by training distribution vs. capability-limited)
Different interpretability signatures (may encode truth internally vs. represent decision to deceive)
The paper provides clear definitions (Sections 3.1-3.2) and honestly discusses boundary cases (Appendix A).

3. Comprehensive and systematic benchmark survey. The review of 35 benchmarks (Tables 6-7, Appendix E) is thorough and well-organized. The coding along four dimensions (object, mechanism, type, audience) enables quantitative gap analysis (Tables 3-4, Section 6).
Key findings:

Object coverage: World/System Claims 46%, Attribution/Provenance only 5.7% (Table 3)
Mechanism coverage: Commission 100%, Omission 14%, Pragmatic Distortion 0% (Table 4)
Type coverage: Behavioral 66%, Strategic 31% (Table 5)
Audience coverage: User 83%, Evaluator 11%, Training Process 5.7%

These statistics clearly identify under-covered areas.

4. The "omission" and "pragmatic distortion" categories are valuable. Highlighting that:

Omission (failing to disclose uncertainty, limitations, counterevidence) is rarely tested (14% of benchmarks)
Pragmatic distortion (technically true but misleading framing) has zero dedicated benchmarks

...is an important contribution. The paper correctly notes (Section 6.2):

"Pragmatic distortion may be particularly dangerous: technically true but misleading statements evade fact-checking, and testing for them requires sophisticated judgment about recipient inferences."

This gap is actionable.

5. Thoughtful risk prioritization (Section 7.3). The five-factor framework (current vs. potential harm, scalability, tractability, reversibility, mechanism neglect) provides structured thinking about which cells deserve attention:
Priority cells identified:

Behavioral: Attribution/provenance × commission (citation fabrication), Belief/uncertainty × omission (failure to express uncertainty), World/system × pragmatic distortion
Strategic: Stated objectives × all mechanisms (alignment faking), Declared capabilities × underclaiming (sandbagging), Future commitments × commission (false promises), All objects × omission

6. Practical contributions: reporting template and recommendations. The minimal reporting template (Appendix H) for new benchmarks is concrete and useful. The recommendations (Section 8) are actionable:

For benchmark designers: prioritize omission, pragmatic distortion, attribution, capability self-knowledge
For evaluators: use incentive-sensitive designs, vary context cues
For developers: monitor citation accuracy, train for calibrated uncertainty

**Weaknesses:**

Weaknesses
1. The behavioral-strategic distinction is presented as clearer than it is in practice.
The paper claims (Section 3):

"These scenarios differ not in their observable output but in the computational process that produced it."

But the distinction requires inferring internal processes that we cannot directly observe. The paper acknowledges (Appendix A) that behavioral and strategic deception "may form a continuum" and provides boundary cases (specification gaming, sycophancy), but the main text presents them as distinct categories.
Problems:

Operational difficulty: How do we actually determine if sycophancy in a specific instance is behavioral (learned disposition) vs. strategic (representing that agreement → positive ratings)? The paper proposes methods (Section 5.3: incentive-sensitive evaluation, process inspection, behavioral consistency, interpretability probes) but acknowledges "each has limitations."
Most benchmarks don't distinguish: Table 8 shows 66% behavioral, 31% strategic, but only 3% "ambiguous." This suggests the survey coded benchmarks as one or the other, but many benchmarks may not actually distinguish—they just measure deceptive outputs without diagnosing mechanism.
The practical value of the distinction depends on having reliable ways to make it. If we cannot reliably distinguish behavioral from strategic in deployed systems, the taxonomy's primary organizing principle has limited practical utility.

Suggested improvement: Be more explicit about when the distinction is clear (e.g., controlled studies like Greenblatt et al. 2024 that vary incentives) vs. when it's ambiguous, and acknowledge that many real-world instances will fall in a grey zone.


2. The "objects of misrepresentation" categories have some overlap and ambiguity.
The seven object categories are:

World/System Claims
Belief & Uncertainty Reports
Reasoning & Justification
Attribution & Provenance
Declared Capabilities
Future Commitments (strategic only)
Stated Objectives (strategic only)

Overlaps/Ambiguities:

Reasoning & Justification vs. Stated Objectives: If a model provides false reasoning about why it chose an action, is this misrepresenting reasoning (behavioral) or stated objectives (strategic)? The paper distinguishes these but the boundary is unclear.
Declared Capabilities vs. World/System Claims: If a model falsely claims "I have access to the internet" (when it doesn't), is this a capability claim or a world/system claim? It's both.
Belief & Uncertainty vs. Reasoning: If a model says "I'm uncertain because..." and provides a false reason for its uncertainty, which category?

Impact: When coding benchmarks or instances, ambiguous cases may be arbitrarily assigned to one category, reducing the taxonomy's reliability.
Suggested improvement: Provide decision rules for ambiguous cases or acknowledge that some instances span multiple categories.


3. The survey methodology is not fully transparent.
The paper states (Section 6):

"We surveyed 35 benchmarks related to deceptive outputs in LLMs and coded each according to four dimensions..."

Missing details:

Selection criteria: How were the 35 benchmarks selected? Was this exhaustive (all benchmarks meeting certain criteria) or convenience sample? If convenience, what's missing?
Coding process: Who coded the benchmarks? How many coders? What was inter-rater reliability? For dimensions with subjective judgments (especially behavioral vs. strategic), agreement rates would strengthen credibility.
Inclusion/exclusion: Were there borderline cases excluded? If so, on what grounds?

Impact: Without these details, we cannot assess whether the gap analysis is comprehensive or whether it might reflect sampling bias.


4. The "pragmatic distortion has zero benchmarks" claim may be too strong.

Summarization faithfulness research (e.g., Wang et al. 2020, cited in Section B.1) evaluates whether summaries mislead readers even when factually consistent. The paper acknowledges this but dismisses it as "primarily operationaliz[ing] the problem as factual inconsistency (commission)."
Selective presentation in argumentation: Work on one-sided arguments or cherry-picked evidence (not cited) could fall under pragmatic distortion.
Implicature and pragmatics in NLP: There's a literature on conversational implicature (Grice's maxims) and how language can mislead through implicated meaning while being literally true.

Alternative interpretation: Perhaps pragmatic distortion is under-studied but not "entirely neglected"—it may exist under different terminology (e.g., "faithfulness," "balanced presentation," "implicated meaning").

5. The paper lacks empirical validation of the taxonomy's utility.
The taxonomy is presented as a conceptual framework, but the paper does not demonstrate that:

Independent coders agree on classifications: Can two researchers independently code the same instance and agree on object, mechanism, and type?
The taxonomy guides mitigation: Does knowing a behavior is "behavioral omission of uncertainty" vs. "strategic commission of false facts" lead to different (and effective) interventions?
The categories are exhaustive: Are there important deceptive behaviors that don't fit the taxonomy?

---

### Official Review · Reviewer_b1Gi · 2026-02-22
**A timely and conceptually strong contribution that meaningfully advances fairness-aware evaluation in alignment and agentic systems, though methodological transparency and minimal operational instantiations would strengthen its impact.**

**Rating:** 3
**Confidence:** 4

**Summary:**

This paper proposes a unifying taxonomy of deceptive behavior in large language models across three dimensions: behavioral vs. strategic deception, object of misrepresentation, and mechanism (commission, omission, pragmatic distortion). By coding 35 existing benchmarks, the authors identify systematic coverage gaps, particularly in omission, pragmatic distortion, attribution/provenance, and evaluator-directed deception and provide recommendations for incentive-sensitive, audience-aware evaluation. The work aims to unify fragmented literatures and guide future fairness and alignment benchmarking.

**Strengths:**

- Clear and operational taxonomy that bridges hallucinations, sandbagging, alignment faking, and strategic deception.
- Systematic benchmark gap analysis highlighting under-explored mechanisms relevant to fairness and alignment.
- Strong relevance to alignment procedures and evaluation design, especially in incentive-sensitive settings.

**Weaknesses:**

- Survey methodology for the 35 benchmarks lacks transparency (inclusion criteria, coding reliability).
- No concrete operational benchmark or protocol for high-priority gaps (e.g., pragmatic distortion).
- Attribution/provenance coverage assessment may understate existing grounding/evidence-based work.

---

### Meta-Review · Area_Chair_nm3A · 2026-02-27

**Recommendation:** Reject
**Confidence:** 5

**Metareview:**

This paper proposes a unified taxonomy for classifying deceptive behaviors in large language models, organizing phenomena such as hallucinations, sycophancy, citation fabrication, and alignment faking into a common framework.  By coding 35 existing benchmarks, the authors identify systematic coverage gaps, particularly in omission, pragmatic distortion, attribution/provenance, and evaluator-directed deception, and provide recommendations for incentive-sensitive, audience-aware evaluation.

The reviewers found this to be an interesting direction; however, there is a notable lack of clarity in the presentation of the results and methodology. The submission would benefit from improved presentation to enable discussion and understanding of the contribution.

---

### Decision · Program_Chairs · 2026-03-02

Reject